# The Effect of Bee Venom and Melittin on Glioblastoma Cells in Zebrafish Model

**DOI:** 10.3390/molecules30153306

**Published:** 2025-08-07

**Authors:** Agata Małek, Maciej Strzemski, Jacek Kurzepa

**Affiliations:** 1Department of Medical Chemistry, Medical University of Lublin, 20-093 Lublin, Poland; 2Department of Analytical Chemistry, Medical University of Lublin, 20-093 Lublin, Poland; maciej.strzemski@umlub.edu.pl

**Keywords:** glioblastoma, bee venom, melittin, zebrafish model

## Abstract

Previous in vitro studies have shown the therapeutic potential of bee venom (BV) against different types of glioblastoma cells. Our aim was to evaluate the cytotoxic effect of BV on glioma in the zebrafish model. First, safe concentrations of BV and melittin were determined by determining the LD_50_ for each substance. Two human glioma cell lines, 8MGBA and LN-229, were used in this study. After staining the tested cells for visualization under UV light, they were then implanted into 2-day-old zebrafish embryos. Zebrafish were incubated for 3 days with crude BV and melittin at concentrations of 1.5 and 2.5 µg/mL vs. control group. Tumor growth was assessed with a stereo microscope. We found differential proliferative responses of two human glioma lines in a zebrafish model. The 8MGBA cell line, but not LN-229, showed proliferative potential when implanted into 2-day-old zebrafish embryos. This study showed a dose-dependent cytotoxic effect only for BV against 8MGBA cells. The observed cytotoxic effect is not dependent on the presence of the peptide melittin—the main BV component with the greatest cytotoxic potential. Simultaneously, a slight increase in LN-229 cell proliferation was observed after 3 days of incubation with melittin at a concentration of 2.5 µg/mL. This indicates that any consideration of bee venom as a therapeutic substance must take into account the type of glioblastoma.

## 1. Introduction

The glioblastoma multiforme is the most frequent and aggressive malignant tumor of the central nervous system, with a median survival after progression ranging from 6 to 9 months [1]. It originates from astrocytic glial cells and is characterized by infiltrative growth and abundant vascularization [2]. The current standard of treatment for patients with newly diagnosed glioblastoma includes the maximal safe resection, which is followed by radiotherapy and temozolomide (TMZ) chemotherapy [1]. The blood–brain barrier is considered a major limiting factor in the effectiveness of chemotherapy. Temozolomide, due to its small size and lipophilic properties, has 100% bioavailability after oral administration, rapid absorption, a good biodistribution, and the ability to cross the blood–brain barrier. However, its tumor tissue levels represent at most only 20% of systemic levels of the drug [3].

Another factor indicated as an impediment to effective treatment is the heterogeneity of glioblastoma, both between patients and at the intratumor level [4]. Genetic mutations and the diverse cellular composition within the tumors, as well as the presence of glioblastoma stem cells, are responsible for treatment resistance and high tumor recurrence rates [5,6]. A multiple-drug approach, such as the combination of temozolomide and thalidomide, has been demonstrated to have better efficacy in patients with glioblastoma multiforme compared to monotherapy [7,8]. The combination of natural extracts with conventional chemotherapy is particularly interesting because of the chance to increase not only the effectiveness but also the safety of the treatment [9].

Natural compounds can be valuable therapeutic tools in the treatment of cancer, as exemplified by the commonly used taxanes, a group of cytostatic drugs isolated from the bark of yew trees (*Taxus* spp.) [10]. Animal toxins can also be a valuable source of substances useful in the treatment of various diseases, and among them bee venom is distinguished by its multidirectional health-promoting effects. Honey bee venom (BV, apitoxin) is a complex mixture of many constituents with a fluctuating composition depending on various factors, such as the strain, age, social position of the bees, their geographical location, or the season [11]. Bee venom contains protein components such as melittin, apamin, adolapin, procamin A and B, tertiapin, secapin, and mast cell degranulating (MCD) peptide. Among the enzymes found in bee venom are the following: phospholipase A_2_ (PLA_2_), phospholipase B (PLB), hyaluronidase, lysophospholipase, acid phosphomonoesterases, as well as α-glucosidase. Apitoxin is also abundant in physiologically active amines and neurotransmitters such as histamine, dopamine, and noradrenalin [12,13].

The main component of apitoxin is the polypeptide melittin, which represents approximately 40–50% of the dry weight of the venom [14]. It is a water-soluble, amphipathic linear polypeptide consisting of 26 amino acids, with a molecular weight of 2840 Da and the chemical formula C_131_H_229_N_39_O_31_ (Figure 1) [11].

Melittin is a natural detergent with high surface and membrane tension. It has the ability to incorporate into the cell membrane and form ionic pores leading to disruption of the phospholipid bilayer structure. The advantage of lytic peptides is that they selectively disrupt the membrane structure of cancer cells, because compared to normal cells, which have a low membrane potential, the membrane of neoplastic cells has a higher membrane potential. Melittin tetramers also induce depolarization of nerve endings and trigger pain [12,15]. Melittin is a compound which increases capillary permeability, reduces blood coagulation, stimulates smooth muscle, and may act as an immunostimulant or immunosuppressant. Higher doses of melittin exert inflammatory and hemolytic effects [14,16].

The subsequent main component of bee venom, phospholipase A_2_ (PLA_2_), represents 10–12% of its dry weight and is known as the main allergen of apitoxin, which can induce anaphylactic shock. PLA_2_ catalyzes the hydrolysis of the sn-2 ester bond of glycerophospholipids to release free fatty acids and lysophospholipids. At the cellular level, PLA_2_ modulates the release of arachidonic acid and the production of eicosanoids, which are strong mediators of inflammation [17]. It is worth mentioning that melittin can also increase PLA_2_ activity and therefore affect cells [11].

Another peptide found solely in bee venom is apamin, a neurotoxin containing 18 amino acid residues linked by two disulfide bridges. Apamin represents approximately 2–3% of the dry weight of bee venom and irreversibly blocks Ca^2+^-activated K^+^ channels (SK channels) in the central nervous system [18]. Apamin is considered a promising neurotherapeutic agent that acts by modulating the neuroinflammatory response, which makes it a candidate for treating a wide range of neurological diseases. Studies on the use of apamin in the treatment of Alzheimer’s disease have demonstrated its potential to increase neuronal excitability, synaptic plasticity, and organ potentiation in the hippocampus by blocking SK channels. However, such a therapeutic approach requires the use of optimal doses of apamin that are low enough to avoid neurotoxicity. In addition, apamin has a protective effect on dopamine neurons, so it can be used in the treatment of Parkinson’s disease, as well as in the regulation of synaptic plasticity and memory [19,20,21].

Despite the proven anti-inflammatory, antimicrobial, and antiviral effects of melittin, its anticancer activity is especially significant for clinical applications [14,22]. Numerous studies have shown that both BV and melittin can exert noticeable toxic effects in various cancer cells, such as lung, breast, bladder, prostate, liver, kidney, and leukemia cancers, among others, with significantly lower effects on normal cells [12]. Melittin has been demonstrated to induce apoptosis in various cancer cell lines. In addition, it affects cell proliferation, necrosis, and angiogenesis, as well as the motility, migration, metastasis, and invasion of tumor cells [23]. The mechanisms of melittin’s actions are complementary, which makes melittin an extremely interesting anticancer substance. Melittin has also been demonstrated to affect signal transduction and various regulatory pathways. For example, melittin inhibited the JAK2/STAT3 pathway, induced NF-κB inactivation, and affected the matrix metalloproteinase pathway and pathways mediated by caspases [14].

Animal models are definitely an imperative tool for studying the complex interactions between neoplastic cells and the tumor microenvironment as well as mechanisms of metastasis. The zebrafish model has become a powerful instrument for studying many human diseases, including cancer biology, and helping to explore the efficacy of potential anti-cancer substances within a short period of time.

The zebrafish (*Danio rerio*) is a broadly used vertebrate model for in vivo studies due to its specific characteristics. This species exhibits ex utero development, which allows the manipulation of organisms at early stages of embryonic development [24]. In addition, 71.4% of human genes were shown to have at least one ortholog of zebrafish, while 82% of human disease-related genes have at least one ortholog of *D. rerio* [25]. The transparency of *D. rerio* tissues allows in vivo imaging of fluorescently labeled cells in real-time [24]. Moreover, in the zebrafish, temporary separation occurs of the maturation of the innate and adaptive immune systems. The innate immune system is functional at 2 days post fertilization (dpf), while the adaptive immune system does not mature until 28 dpf. The lack of adaptive immunity in zebrafish larvae makes it a suitable in vivo model for injecting cancer cells due to the elimination of rejection risk and the possibility of survival of cancer cells in the larvae, as well as attacking surrounding tissues and forming metastases [26].

The aim of this study was to evaluate the anticancer properties of bee venom vs. its main component, melittin, on glioblastoma cells in the *D. rerio* model.

## 2. Results

### 2.1. Estimation of LC_50_ and Assessment of the Potential Cardiotoxicity of the Substances

The wild-type zebrafish embryos at 2 days post fertilization (dpf) were treated with crude bee venom and melittin at ten different concentrations ranging from 0.5 to 10 μg/mL for 72 h to evaluate the toxicity levels of the tested substances. During the experiment, the embryos developed and became larvae at 72 h post fertilization. Therefore, in this article, we use the term “embryo” to refer to the 2-day-old zebrafish at the beginning of the experiment. In turn, we use the term “larvae” for further stages of the experiment when the zebrafish had reached or exceeded the age of 72 h post fertilization. Both crude bee venom and melittin demonstrated a decrease in the viability of zebrafish larvae. The LC_50_ values, indicating the concentration at which 50% of the zebrafish larvae are no longer viable, were slightly lower for melittin (3.308 µg/mL) than for crude bee venom (3.755 µg/mL). During the experiment, no significant increase in the frequency of malformations was observed in the larvae exposed to the tested substances, and heartbeat observations did not show their cardiotoxic properties (Figure 2). Based on the toxicity results, two safe doses of bee venom and melittin (1.5 µg/mL and 2.5 µg/mL) were selected for further experiments on zebrafish.

### 2.2. Assessment of the Antiglioblastoma Activity of Bee Venom vs. Melittin

To evaluate the potential anticancer effects of bee venom and melittin on glioblastoma multiforme cells in vivo, a model of *D. rerio* was used. The cells of two glioblastoma lines were injected into the yolk sacs of 2 dpf zebrafish embryos, which were then exposed to two different concentrations (1.5 µg/mL and 2.5 µg/mL) of bee venom and melittin to evaluate their antiproliferative potential (the control group was not treated with any substance). The viability of glioblastoma cells was measured after 72 h of larvae incubation with the tested substances. Representative images of glioblastoma cells’ xenotransplantation into zebrafish embryos are provided in Figure 3.

Groups treated with crude bee venom showed different responses depending on the type of glioblastoma cells injected into the embryos. A nearly 2-fold increase in tumor volume after 72 h was noted for line 8MGBA in control zebrafish larvae. The bee venom-treated group previously injected with the 8MGBA line showed a noticeable reduction in glioblastoma cell growth compared to the control, and this effect was even slightly greater with the higher BV dose (2.5 µg / ml). However, in zebrafish injected with the 8MGBA cell line and treated with melittin, there was a lack of tumor growth-limiting effect (Figure 4).

The other tested line, LN-229, showed no proliferative potential under in vivo conditions in control embryos. Similar or even higher growth of tumor cells compared to the control group was observed for the group treated with bee venom injected with the LN-229 line. Finally, the group treated with melittin at the tested doses also showed similar or higher growth of glioblastoma cells compared to the control, indicating its limited or lack of inhibitory effect on tumor cell proliferation (Figure 5).

The cell lines used in this study showed varying proliferative potential in the in vivo model, and an antitumor effect was demonstrated only for the actively proliferating line incubated with bee venom. Only in the case of the actively proliferating line (8MGBA) did bee venom show the potential to reduce tumor volume growth, and this effect was more significant at the higher dose. It is noteworthy that melittin showed slightly higher toxicity to zebrafish larvae, which, superimposed with the tumor cell injection loading factor, may have affected the model organism more than the larvae incubated with crude bee venom (Figure 2A,C).

## 3. Discussion

In an era of unprecedented development of pharmacology, increasingly based on biological drugs and genetic preparations, the search for new natural substances with medicinal properties may seem outdated. However, the huge advantage of natural substances over newly synthesized drugs is often thousands of years of human contact with these compounds. Such a period can be compared to the endless Phase IV of clinical trials, meaning many beneficial and unfavorable properties of such substances are well known.

One of these substances is BV. Human contact with this venom is inevitable when collecting honey from wild bees, as evidenced by cave paintings depicting honey collection [27]. Bees, like other animals, use venom to defend themselves against opponents. This compound has a destructive effect on the tissues of the attacker, along with causing pain and other unpleasant sensations, discouraging them from continuing the attack [28].

BV exerts cytotoxic properties against many types of cells. This is an understandable effect due to the role it plays in nature. The main cytotoxic component of BV is melittin. Previous studies have shown that this peptide leads to damage of the cell membrane, initiating the apoptosis pathway in cells. Melittin is also the main compound responsible for hemolysis of red blood cells after entering the bloodstream (see [11] for review). However, the cytotoxic properties of melittin can potentially be used in the treatment of various types of cancers, including one of the most serious cancers—glioblastomas. Previous in vitro studies indicate cytotoxic effects of melittin on different types of glioblastoma cells. In the presence of melittin and crude BV, Hs683, T98G, and U373 cell lines showed signs of late apoptosis and necrosis after only 1 h of incubation [29]. Moreover, melittin at a dose of 2.5 µg/mL enhanced the cytotoxic effect of cisplatin against the glioblastoma line DBTRG-05MG [30]. Our previous in vitro studies have shown that both crude BV and the extracted venom fraction containing melittin have a pronounced cytotoxic effect against human glioblastoma lines compared to physiological cells. The remaining BV fractions, without melittin, showed no or very weak cytotoxic activity [31]. This prompted us to conduct further studies, on the model of *Danio rerio*. This model has been used in research for many years. Its advantages is in the transparency of the organism, rapid development, and ease of genetic manipulation, which make the zebrafish an excellent model for isolated research of various mechanisms, including cancer cell growth [32].

This study began by determining a safe dose of the test substances (BV and melittin), so that their concentration would not kill the larvae before the end of the experiment. Then, we selected two safe concentrations (below LD_50_), in which the embryos with implanted glioblastoma cells were then incubated for 3 days. On the fifth day, tumor cell growth was analyzed in comparison with the beginning of the experiment. Since the LD_50_ for both bee venom and melittin was over 3 µg/mL, we set the doses of both substances used in the experiment at 1.5 and 2.5 µg/mL. In the case of implantation of 8MGBA cells into two-day-old embryos, a significant increase in the number of tumor cells by about 80% was observed on the 3rd day of the experiment in the control group (without incubation with any tested substance). At the same time, a dose-dependent reduction in the number of 8MGBA cells was observed during incubation with BV. Such a relationship was not observed when larvae were incubated with the melittin standard, which suggests that this peptide was not responsible for the antiglioblastoma effect of BV observed in this model. Interestingly, our previous in vitro studies suggested that melittin was responsible for the antiglioblastoma effect in two tested human glioma lines, LN-229 and LN-18 [31]. Currently, in the zebrafish model, in which cells are implanted into living organisms, the effect exerted by the tested substances seems to be different from that in studies conducted on pure lines in cell culture conditions. Another effect was observed in the case of incubation of zebrafish larvae implanted with LN-229 cells, in which we did not observe an increase in the number of tumor cells during the incubation period in the control group as well as in the case of incubation with both concentrations of BV. In the case of incubation with melittin, a small, but statistically significant, unexpected increase in the number of glioblastoma cells was observed at a melittin concentration of 2.5 µg/mL. An analogous result was obtained in two independent repetitions. Although both cell lines are derived from human glioblastoma, their different responses to BV and melittin in the zebrafish model are intriguing. Both lines 8MGBA (CVCL_1052) and LN-229 (CVCL_0393) were derived from glioblastoma cells of adult Caucasian women over 50 years of age. Despite this, the glioblastoma cells differ in molecular subtypes, types of mutations, and profiles of proteins responsible for drug resistance, which directly affects the ability to respond to a specific therapy [33]. Interestingly, BV studied mainly for its cytotoxicity against various cancer cells turned out to be a compound stimulating division and growth in some lines. Kim et al.’s study showed in vitro and in vivo that BV accelerated the growth of adipose-derived stem cells (ASCs) by inducing growth factors, fibroblast growth factor (FGF)-1 and -6, endothelial cell growth factor, and platelet-derived growth factor (PDGF)-C [34]. This observation indicates that BV has proliferation-stimulating properties, although the study did not use cancer cells. In our experiment, both incubation with BV and melittin increased the number of cells on day 3 of the experiment, but only for melittin was this difference statistically significant (Figure 5B). However, to the best of our knowledge, there have been no reports in the literature on the positive effect of the low doses of BV on the growth of cancer cells. Previous reports indicated a situation in which the implanted cancer cells did not multiply or multiplied in a statistically insignificant way in the larva. Studies by Wang et al. showed an increase in the proliferation of pancreatic cancer cells of the BxPC3 line by only about 7% [35]. In another study, physiological BJ cells did not grow or disseminate, suggesting that proliferation is a consequence of the tumorigenic behavior of transformed cells [36].

Finally, it is worth noting the limitations of this study. Obtained results should be considered the beginning of a long journey toward assessing the effects of bee venom on glioblastoma treatment in humans. They cannot be directly applied to the potential effects of BV in humans due to obvious differences between the two species. First, this study only included observations for three days. This was dictated by the zebrafish research model used, which assumes research up to five days of life (between three and five days of age). Therefore, the long-term effects of BV on zebrafish are unknown. Second, BV in humans has numerous adverse effects that complicate its use. Furthermore, BV penetration through the blood–brain barrier (in the case of a tumor, the blood–tumor barrier) is difficult and unpredictable. Therefore, in the treatment of glioblastoma, it is possible to consider administering bee venom directly to the tumor or using various delivery vehicles (e.g., nanoparticles).

A PubMed search for “zebrafish”, “larvae”, and “glioblastoma” in the title/abstract yields only 10 publications from 2012 to 2024. None of them concern the study of BV. This indicates that the research topic undertaken is new, and this publication is the first to present the effect of bee venom and melittin on glioblastoma cells in the zebrafish larvae model.

## 4. Materials and Methods

### 4.1. Cell Culture

Two human glioblastoma cell lines, LN-229 and 8MGBA, obtained from the American Type Culture Collection (ATCC), were used for injections of *D. rerio* embryos. The LN-229 line was maintained in Dulbecco’s Modified Eagle Medium (DMEM), while 8MGBA cells were cultured in Eagle’s Minimum Essential Medium (EMEM), both supplemented with 10% fetal bovine serum, penicillin (10,000 U/mL), and streptomycin (10 mg/mL). The cells were incubated at 37 °C in a humidified atmosphere containing 95% air and 5% CO_2_. Cell cultures were maintained in the logarithmic growth phase by regular passage at 80% confluence.

### 4.2. Zebrafish Husbandry

All experiments with embryos and larvae of *D. rerio* were performed at the Experimental Medicine Center of the Medical University of Lublin, Poland. Through the natural spawning process of the wild-type fish housed at the Zebrafish Facility of the Experimental Medicine Center, fertilized zebrafish eggs were obtained, according to established standard breeding protocols under Specific Pathogen-Free (SPF) conditions. The zebrafish used to spawn were maintained under a 12 h light/12 h dark light cycle at 28.5 °C; pH 7 (± 0.5).

### 4.3. Drug Toxicity

To determine the toxicity of the tested substances—commercially available, dried bee venom powder obtained from *Apis mellifera* “https://www.beevenomlab.com (accessed on 1 Jun 2025)” detailed chromatographic analysis of the bee venom used in this study can be found in the Appendix A) and a melittin standard (Sigma-Aldrich, Saint Louis, MO, USA, Cat. No. 4446605)—the zebrafish model was used. Viable 2 dpf embryos were previously dechorionized and examined using an optical microscope to select individuals without any visible malformations. E3 solution (5 mmol/L NaCl, 0.17 mmol/L KCl, 0.33 mmol/L CaCl_2_, and 0.33 mmol/L MgSO_4_, without methylene blue and with a pH value of approx. 7.2) was used both as zebrafish culture medium and as the solvent for preparing solutions of the tested substances. The bee venom and melittin stock solutions of 1 mg/mL in E3 medium were prepared and vortexed for 1 min directly before each experiment. The stock solutions were then diluted in E3 to obtain the required concentrations for zebrafish incubation. Then, selected zebrafish were exposed to solutions of tested substances for up to 72 h at concentrations from 0.5 to 10 μg/mL. The experimental system was static, in order to ensure that changes in the concentrations of the solutions did not exceed the range of 20% of the nominal concentration values. The experiment was conducted in 24-well Corning^®^ Costar^®^ TC-Treated plates (Sigma Aldrich, St. Louis, MO, USA) by placing five embryos per well and ten per group. Three independent replicates were performed for each compound. Plates were covered and stored in an incubator set at 28 ± 0.5 °C in the 12 h/12 h light/dark mode. The toxicity was evaluated based on the zebrafish lethality and the presence of malformations at the end of the 72 h exposure period (when the zebrafish were no more than 5 days old). In addition, the heart beats per minute were measured for viable larvae in both groups: control and incubated with the tested substances. The LC_50_ value of each analyzed substance was calculated. The aim was to select simultaneously safe and maximally high therapeutic doses for further experiments.

In accordance with Directive 2010/63/EU concerning the protection of animals used for scientific purposes, the larvae of zebrafish are classified as self-feeding up to 120 hpf (first 5 days of life). Therefore, they are not subject to the same regulations as mature animals, and approval from the Local Ethical Committee is not required.

### 4.4. Zebrafish Xenotransplantation

LN-229 and 8MGBA glioblastoma cells were stained with Invitrogen™ Vybrant™ CM-DiI Cell-Labeling Solution (Thermo Fisher Scientific, Waltham, MA, USA) following the manufacturer’s protocol. *D. rerio* embryos 2 days after fertilization were injected into the yolk sac with 500 labeled cells of a particular glioblastoma cell line. Tumor cells were injected using a Narishige IM-300 microinjector (BioMedical Instruments, Zöllnitz, Germany) and glass microinjection needles without filaments (World Precision Instruments, Sarasota, FL, USA). Injected embryos were incubated at 28.5 °C for 1 h, and then photographs were taken of all embryos selected for the experiment (baseline). During the injection of tumor cells and for the purpose of taking photos, zebrafish were anesthetized with 4 mg/mL of 3-aminobenzoate ethyl ester methanesulfonate salt (tricaine, Sigma-Aldrich, cat # A-5040, St. Louis, MO, USA) in E3 medium. Injected zebrafish were transferred to 24-well plates and incubated in E3 (control group) or E3 solution containing the analyzed substances (treatment groups). Two concentrations of bee venom and melittin were used in the experiment—1.5 and 2.5 µg/mL—and the larvae were incubated at 32 °C for 72 h. The second series of images of injected larvae was taken after 72 h to compare changes in tumor size. Tumor growth was assessed with a SteREO Discovery.V8 stereo microscope for fluorescence studies (Carl Zeiss Microscopy, Jena, Germany) using a 3.2x objective. The obtained data were analyzed using ImageJ software, version 1.53e, the National Institute of Health, USA “https://imagej.net/ij/ (accessed on 1 June 2025)”.

### 4.5. Statistics

The results of cell viability implanted into zebrafish larvae were expressed as mean and SEM. LD_50_ values for the tested substances were assessed using GraphPad Prism software (version 10, Boston, MA, USA). The statistical significance of differences between the beginning of the experiment and 72 h later was assessed using one-way ANOVA followed by Dunnett’s post hoc test (https://www.statscalculators.com, accessed on 29 June 2025). Values were considered significant if *p* < 0.05.

## 5. Conclusions

This study indicates a selective, cell line-dependent effect of bee venom on the growth of glioblastoma cells in the zebrafish model. The observed cytotoxic effect is not dependent on the presence of melittin—the BV component with the greatest cytotoxic potential. This indicates that any consideration of bee venom as a therapeutic substance must take into account the type of glioblastoma.

## Figures and Tables

**Figure 1 molecules-30-03306-f001:**
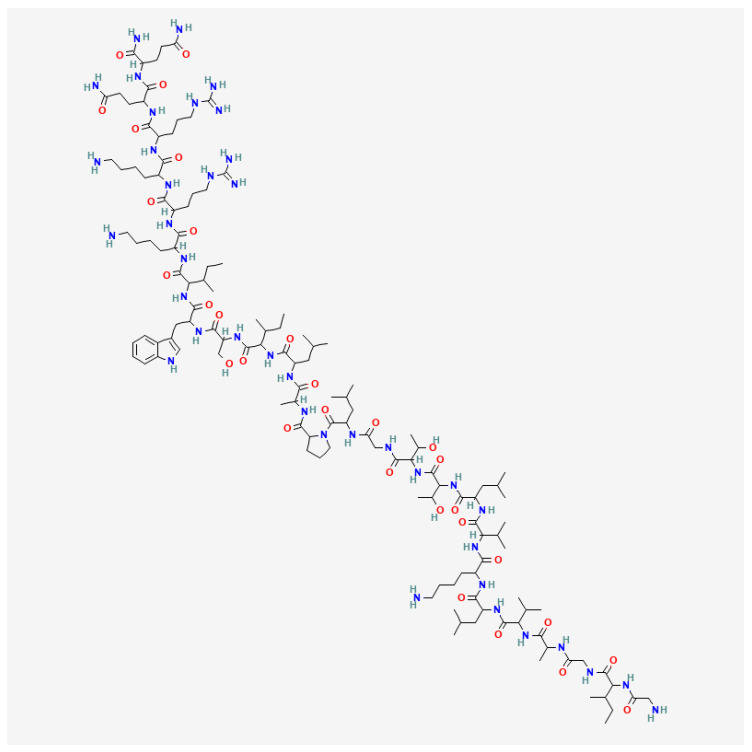
Structural formula of melittin (PubChem CID 16129627).

**Figure 2 molecules-30-03306-f002:**
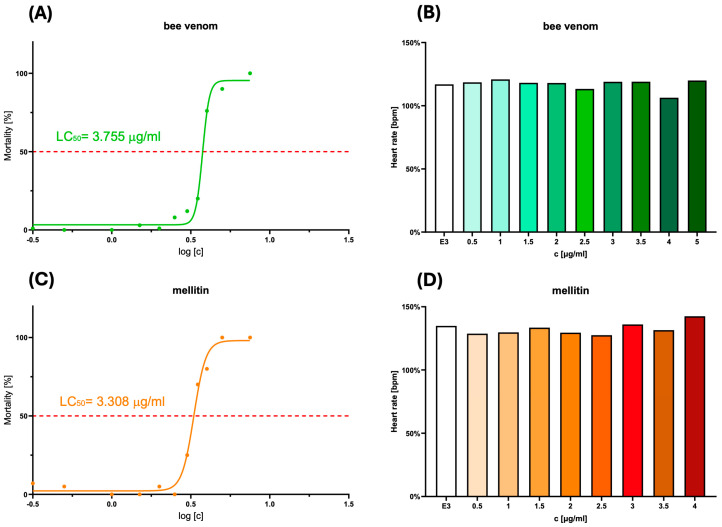
The mortality of zebrafish exposed to different concentrations of BV (**A**) and melittin (**C**). The percentage of zebrafish mortality (*y*-axis) as a function of the logarithm of the drug concentration (*x*-axis). The red dashed line represents the 50% viability, which is used to determine the LC_50_ (Lethal Concentration 50) value for each compound. The LC_50_ value is the concentration at which 50% of the zebrafish larvae are expected to die. The right side of the figure indicates the lack of cardiotoxicity of the bee venom (**B**) and melittin (**D**) concentrations used. The mean heart rate did not change significantly.

**Figure 3 molecules-30-03306-f003:**
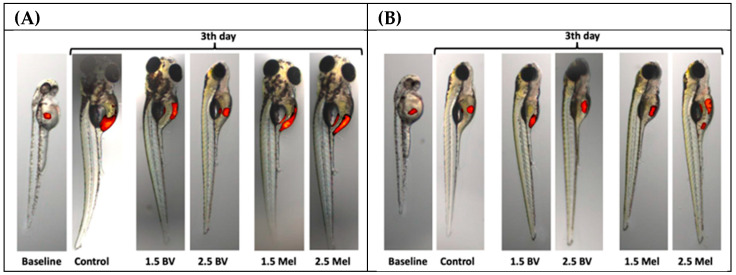
Representative images of glioblastoma cells’ xenotransplantation into zebrafish embryos. Microscopic observation of the 8MGBA cells (**A**) and LN-229 cells (**B**) at 5 dpf zebrafish xenograft upon treatment with BV and melittin (Mel) (1.5 µg/mL and 2.5 µg/mL) vs. control zebrafish embryos xenotransplanted with glioblastoma cells and incubated without BV or Mel.

**Figure 4 molecules-30-03306-f004:**
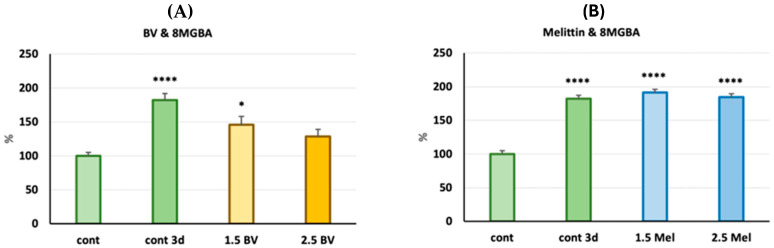
Bar graphs representing the percentage of 8MGBA cells xenotransplanted into zebrafish embryos treated for 72 h with BV (**A**) or melittin (**B**) at the concentrations of 1.5 µg/mL and 2.5 µg/mL for each compound, or without any treatment (cont 3d). Initial measurement of cell percentage before treatment was set to 100% (cont). Mean (SEM), * *p*  <  0.05, **** *p*  <  0.0001 compared to control.

**Figure 5 molecules-30-03306-f005:**
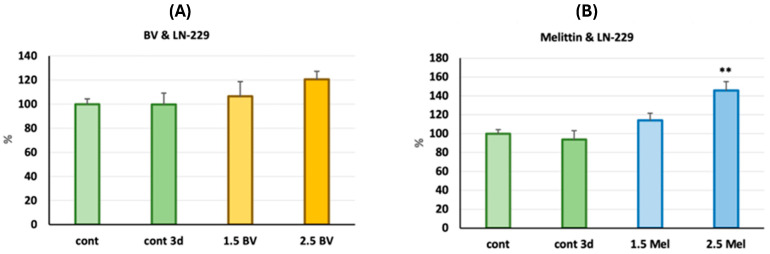
Bar graphs representing the percentage of LN-229 cells xenotransplanted into zebrafish embryos treated for 72 h with BV (**A**) or melittin (**B**) at the concentrations of 1.5 µg/mL and 2.5 µg/mL for each compound, or without any treatment (cont 3d). Initial measurement of cell percentage before treatment was set to 100% (cont). Mean (SEM), ** *p* < 0.001 compared to control.

## Data Availability

Inquiries can be directed to the corresponding author.

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
