# Peer review of "The Effect of Bee Venom and Melittin on Glioblastoma Cells in Zebrafish Model"

_molecules, 2025, doi:10.3390/molecules30153306_

Round 1
Reviewer 1 Report
Comments and Suggestions for Authors
This is an interesting paper about the effect of bee venom on GBM cells in zebrafish model. Although there seems to be as yet a long way to go, the present study could be an important step forward to a better treatment strategy for GBM. I have a few comments about the study.
#Title
Since melittin is also an important participant in this paper, it should be incorporated into the title in some way or other. For example, ‘The effect of bee venom vs. melittin on…’.
#Abstract
-larvae vs embryos: Larvae and embryos are the same or different? Larvae are those after birth (embryos)? Not only in the abstract but also elsewhere in the MS, larvae and embryos seem to alternate with each other without any rules. This makes this MS somewhat difficult to understand, particularly for unfamiliar readers. If larvae and embryos are to be used in this paper, they should be used more strictly. I could not understand why larvae in here and why embryo in there.
-Melittin: What is melittin? What kind of substance is it? A brief mention about it is warranted for unfamiliar readers. This is because an abstract needs to be self-sufficient.
#Introduction
-Melittin: According the description here, melittin sounds quite important to understand this paper. The 2nd and 3rd paragraph delineate it, but just words or sentences cannot convey the general image of it to the readers. How about inserting representative tables or figures showing its chemical (structural formula, etc.), physiological, or therapeutic profiles, thus far reported, in addition to its relation to BV. This will surely help the readers get prepared to read the following sections of this article.
-After all, is it (Melittin) a component of BV or a substance separate from BV? To understand these points is very important to interpret the Results which showed the discrepancy of the effect on GBM cells between BV and Melittin. To be honest, I could not understand at first why melittin had to participate in this experiment.
-pdf: This derives from ‘days after fertilization’, which is okay, but ‘days post fertilization’ is actually what it literally comes from. Not ‘days after fertilization’ but ‘days post fertilization’ should appear in line 99 for better understanding. (It is actually so in line 106, then, why is it not so in line 99?)
#Results
-Figure 1 B/D: As the proof of absence of cardiotoxicity, heart rate (HR) is used. I admit that HR could be a choice, but is it common to use HR to prove the presence or absence of cardiotoxicity?
-Figure 2/3
> In the legend, it says ‘8MGBA’. But in other parts of the MS, it says ‘8-MG-BA’. Which is correct? An identical thing should be described with an identical form of writing.
> **** or *: This means that ‘**** or *’ is significantly higher than ‘cont’? Probably so, but it would be better if we can see this just by seeing the figures or reading the legends.
-line 167-169: I could not find the results corresponding to this statement. Is it Figure 1D???
#Discussion
It is sometimes unclear to which result does a statement in here corresponds. For example, I could not understand to which result the statement in line 237-239 corresponds. Is it Figure 4B??? It is better to indicate the corresponding result (Figure 4B, etc.) again when making an affirmative statement for easier understanding on the side of the readers.
Reviewer 2 Report
Comments and Suggestions for Authors
Journal: Molecules
Submission ID: 3761014
Paper Type: Article
Title: The effect of bee venom on glioblastoma cells in zebrafish 2 model
In this manuscript, submitted as Article to the Molecules journal, the authors (with previous experience) presented important new data on the effects of the melittin and the bee venom on glioblastoma cells, in vivo. The experimental model used, and the results are original and interesting. The subject is one of actuality both from a clinical and scientific point of view. The topic of the manuscript is relevant to the Molecules and is appropriate for this journal.
The text of the manuscript is cursive, correctly written, in a style that captivates attention. Introduction provides enough background to illustrate the importance of the topic; The study was well designed and conducted; Materials and Methods are appropriate and described in enough detail; Results are clearly presented using 4 illustrative figures, and Conclusions reflect the obtained results. The paper is based on an appropriate number of references, many of them relatively recent or very recent - confirming the actuality of the subject.
Therefore, this reviewer recommends the publication of this manuscript into the Molecules journal, but after a minor revision concerning the following 9 points:
- The authors have to add the methodology in the Abstract;
- Ref [10] is not correctly cited in line 51 (ref [10] is a review about the applications of BV);
- The authors have to clearly state the objective(s) of their study at the end of Introduction;
- In line 126, the authors should introduce another subsection (2.2. Anticancer effect);
- In lines 146-147, the authors should specify “at the tested doses”;
- In line 240, the authors should specify “of the low doses of BV”;
- At the end of Discussion section, the authors should add, as limitation of their study, a small paragraph to compare and discuss the possibility to analyze the effects of the BV on tumors grown for 3 days in their experimental conditions with the possible effects of BV in vivo on tumors grown for several month in the patients’ brain.
- The authors must be consistent when presenting the origin of the devices, reagents and software;
- 10 references have different format.
Reviewer 3 Report
Comments and Suggestions for Authors
the research topic is very interesting and could have high impact on the scientific community. the authors choose well-known fact about potential benefit of bee venom in various diseases therapy especially on cancer patience and investigated applicable dosage without side effects on zebra fish models. the introduction is very informative, pointed out importance of the study and underline key topics necessary for proper understanding of the research. the experimental part explain experiments on zebra fish very precisely but it doesn't give the instruction for preparation of bee venom. the obtained results are presented in logical manner and properly discussed. the conclusion summarize the important findings and underline those who responds to research framework. the number of picture and tables are satisfactory and give additional value to article making it more understandable.
but i have few concerns which authors should explain before article become acceptable for publication,
what is the content of melittin in bee venom? does LC5o value corresponds to it?
explanation of other active components such as apamin and phospholipase should be given!
what was the purity of bee venom?
Round 2
Reviewer 1 Report
Comments and Suggestions for Authors
Good work. I wish this study were of some help for patients with GBM in the future.